# Multi-Observer Study on Diagnostic Accuracy of Pediatric Renal Tumors Imaged with Higher-Harmonic-Generation Microscopy

**DOI:** 10.3390/cancers17101693

**Published:** 2025-05-18

**Authors:** Sylvia Spies, Elina Nazarian, Srinivas Annavarapu, Paola Collini, Aurore Coulomb L’Hermine, Ellen D’Hooghe, Jozef Kobos, Guillaume Morcrette, Mariana A. Morini, Sergey D. Popov, Rajeev Shukla, Isabela Werneck da Cunha, Cornelis P. van de Ven, Marry M. van den Heuvel-Eibrink, Ronald R. de Krijger, Marie Louise Groot

**Affiliations:** 1LaserLab Amsterdam, Department of Physics, Faculty of Science, Vrije Universiteit Amsterdam, 1081 HZ Amsterdam, The Netherlands; e.nazarian@amsterdamumc.nl (E.N.); m.l.groot@vu.nl (M.L.G.); 2Real-World Research Ontada, Boston, MA 02110, USA; 3Soft Tissue Tumor Pathology Unit, Fondazione IRCCS Istituto Nazionale dei Tumori, 20133 Milan, Italy; 4Department of Pathology, APHP, Armand Trousseau Hospital, Sorbonne University, 75012 Paris, France; 5Department of Pathology, Oslo University Hospital, Rikshospitalet, P.O. Box 4950 Oslo, Norway; eldhoo@ous-hf.no; 6Department of Histology and Embriology, Medical University of Lodz, 90-419 Lodz, Poland; 7Department of Pathology, Central Clinical Hospital of the Medical University of Lodz, 92-213 Lodz, Poland; 8Department of Fetopathology, Robert Debré Hospital, Université Paris Cité, APHP, 75019 Paris, France; 9Instituto D’Or for Research and Education (IDOR), São Paulo 05403, Brazil; 10Department of Pathology, Rede D’Or Sao Luiz Hospital, São Paulo 04321-120, Brazil; 11Department of Pathology, Sidra Medicine, Doha P.O. Box 26999, Qatar; 12Department of Paediatric Histopathology, Alder Hey Children’s NHS Foundation Trust Hospital, Liverpool L12 2AP, UK; 13Princess Máxima Center for Pediatric Oncology, 3584 CS Utrecht, The Netherlands; 14Wilhelmina Children’s Hospital, University Medical Center Utrecht, 3584 CX Utrecht, The Netherlands; 15Department of Pathology, University Medical Center Utrecht, 3584 CX Utrecht, The Netherlands

**Keywords:** pediatric kidney tumors, nonlinear microscopy, third-harmonic generation, second-harmonic generation, two-photon excited autofluorescence, Wilms tumor, nephroblastoma, renal cell carcinoma, congenital mesoblastic nephroma, renal pathology

## Abstract

Higher-harmonic-generation microscopy (HHGM) is an innovative imaging technique that enables rapid visualization of tissue without the need for preparation or staining. In this study, we demonstrate that HHGM provides comparable information about the renal architecture to conventional histology. Remarkably, pathologists achieved very high accuracy in distinguishing between normal renal tissue, abnormal renal tissue, and (pediatric) renal tumors in HHGM images. Therefore, HHGM could serve as a powerful tool for the rapid assessment of sample representativeness, making it particularly valuable for biobanking or other analyses that require a high tumor content. In addition, HHGM could serve to provide intraoperative feedback on renal biopsies.

## 1. Introduction

Wilms tumor (WT), also known as nephroblastoma, accounts for approximately 90% of pediatric renal tumors [1]. Congenital mesoblastic nephroma (CMN) is most prevalent in children under the age of two months, whereas renal cell carcinoma (RCC) starts to become more prevalent than Wilms tumor in children over the age of fourteen [2,3,4]. Histologically, Wilms tumors are composed of three components in varying proportions: blastema, epithelium and stroma. The International Society of Paediatric Oncology—Renal Tumor Study Group (SIOP-RTSG) developed the UMBRELLA SIOP-RTSG 2016 protocol (further referred to as the UMBRELLA protocol) in which patients are typically treated with chemotherapy before surgery and biopsies are only recommended in specific cases [5,6]. To achieve reliable and reproducible tumor classification, nephrectomy specimens are evaluated according to a standardized protocol [4]. Tumor classification and treatment are determined by the percentage of chemotherapy-induced changes and the percentages of the three components in the remaining viable component of the resected tumor. The blastemal component is the least differentiated, and blastemal-type Wilms tumors are associated with a poorer prognosis [4,7], making accurate assessment crucial for proper classification. Additionally, pediatric renal tumors often exhibit overlapping cell types and histologic patterns, complicating differential diagnosis when relying on H&E staining alone. This highlights the need for supplementary diagnostic techniques [1].

Given the high heterogeneity of Wilms tumors, tissue samples for biobanking are collected from at least three distinct, macroscopically viable areas within the tumor. The UMBRELLA protocol states that frozen section analysis or touch imprints can be used to assess the tissue’s cellular content, viability, and tumor cell content, ensuring the samples are representative and viable [4].

Higher-harmonic-generation microscopy (HHGM) is an innovative label-free imaging technique that combines third-harmonic generation (THG), second-harmonic generation (SHG), and two-photon excited autofluorescence (2PEF). HHGM allows for the visualization of cells, collagen and autofluorescence in unprocessed tissue, in just a few minutes. Recent studies have applied HHGM to brain, breast, lung and thyroid tissue, demonstrating strong correspondence with conventional histology [8,9,10,11] and assessment of malignancy by artificial intelligence algorithms [12,13].

However, to the best of our knowledge, only one or two combinations of these HHGM imaging modalities have been explored for renal tissue, excluding THG, despite its ability to visualize almost all structural components of the tissue [8,9,10,11]. For example, Strupler et al. demonstrated that SHG can quantify renal interstitial fibrosis [14], while Best et al. showed that SHG can assess collagen density and alignment in renal cell carcinomas to distinguish between low-grade and high-grade tumors [15]. The combination of SHG and 2PEF has been used to image pediatric tumors, but these studies used formalin-fixed and paraffin-embedded tissue [16,17].

In contrast to these studies, we include THG in our approach to visualize cells and cell nuclei, in addition to SHG and 2PEF, enhancing the overall information content and improving the resemblance to histological analysis. Moreover, our use of a transportable HHG microscope and fresh, unprocessed tissue underscores the potential of this technique for real-time imaging in the operating room. The primary aim of our study is to evaluate the accuracy with which an international group of expert pathologists can recognize normal renal tissue, abnormal renal tissue and pediatric renal tumors, including Wilms tumors and other renal tumors, on HHGM images, as compared to conventional histology. We also seek to assess the diagnostic accuracy of HHGM in identifying the different components of Wilms tumors, contributing to more efficient and precise tumor diagnosis.

## 2. Materials and Methods

Twenty-nine freshly excised tissue samples from eighteen pediatric (0–18 years) patients treated for a renal tumor in the Princess Máxima Center and undergoing a (partial) nephrectomy were imaged with the HHG microscope. After HHGM imaging, the samples were processed according to standard histopathological protocols. The HHGM images were compared with hematoxylin and eosin (H&E)-stained sections of the same tissue fragment. A multi-observer study was performed with ten experienced pediatric renal pathologists of the SIOP-RTSG.

### 2.1. Ethics Statement

All material was provided by patients from the Princess Máxima Center for pediatric oncology, who had been treated according to the SIOP-RTSG-2016-UMBRELLA protocol. The research was approved by the Biobank and Data Access Committee of the Princess Máxima Center (PMCLAB2022.297). All included patients underwent treatment at the Princess Máxima Center for pediatric oncology and provided written informed consent for participation in the biobank (International Clinical Trials Registry Platform: NL7744; https://onderzoekmetmensen.nl/en/trial/21619, accessed on 23 July 2024). This research project followed The Netherlands Code of Conduct for Research Integrity and the Declaration of Helsinki. The results of this research were not used for patient diagnosis or treatment.

### 2.2. Sample Handling

Tissue samples were transported from the operating room to the laboratory for pediatric oncology of the Princess Máxima Center in Utrecht, The Netherlands. One of the pathologists, who did not participate in the HHGM assessment, selected (vital) tumor and normal samples for HHGM imaging, based on macroscopic evaluation. There was no case selection, only a sample selection from the surgical specimen to obtain (suspected) tumor and, if possible, normal tissue samples. The samples, maximally measuring 10 mm × 10 mm × 5 mm, were placed in a sample holder (µ-dish 35 mm, high glass bottom, ibidi GmbH, Gräfelfing, Germany), with the same orientation as required for histology. HHGM imaging was performed on the bottom layer of the sample, around 20 microns from the glass interface. After imaging, the sample was processed following standard histopathology procedures. Hematoxylin and eosin (H&E) sections were made from the same cutting plane as the HHGM image to have the best correspondence. The gold standard diagnosis was considered to be the diagnosis made by the pathologists of the Princess Máxima Center, based on the histology of the entire tumor, and additional techniques such as immunohistochemistry and molecular techniques. The HHGM images were not involved in the diagnosis.

### 2.3. Image Acquisition

The set-up is shown in Figure 1A,B. A transportable HHG microscope (co-developed with Flash Pathology B.V., Amsterdam, The Netherlands) was used to acquire the HHGM images. A brief explanation of the physical mechanism of HHGM is given in Appendix B. The HHG microscope was similar to that described previously [10,11], with minor modifications. A sub-80fs laser source, centered at 1060 nm (Biolit 2 with precompensation, Litilit, Vilnius, Lithuania), was used to generate the nonlinear signals. The microscope was equipped with an acousto-optic modulator (AOM) to select bunches of 5–10 pulses at a repetition rate of 1 MHz out of the 15 MHz pulse train, to achieve a low average power of 5 mW, with a pulse peak energy of 5 nJ. This ensured a sufficient peak power to generate the nonlinear optical signals while at the same time a low average power to avoid damage (heating) to the tissue. The laser beam was focused using an oil-immersion microscope objective with a high numerical aperture (40x/1.3NA, Nikon, Tokyo, Japan) to obtain a sub-micrometer focus of 0.4 × 0.4 × 2.4 μm^3^ [10]. The sample dish was placed in a sample holder above the objective. Analogue photo-multiplier tubes (H10721-210 and H10721-20, Hamamatsu Photonics, Hamamatsu, Japan) collected the third-harmonic generation (THG), second-harmonic generation (SHG), and two-photon excited autofluorescence (2PEF) signals. Dichroic mirrors and interference filters were used to separate the three signals, resulting in the following detection bandwidths: THG 335–376 nm, SHG 505–545 nm and 2PEF 573–642 nm. Two-dimensional images were acquired by bidirectional raster scanning of the galvanometer mirrors. Larger 2D images were created as a mosaic of smaller images by moving the sample in the x- and y-directions (XYZ stage, Applied Scientific Instrumentation, Eugene, OR, USA) after each small 2D image. Mosaic scanning of preset dimensions was automatically performed with a LabVIEW program (Flash Pathology B.V., version 2.1). In addition, a multi-axis stage controller (Applied Scientific Instrumentation) with a joystick (x,y-direction) and control wheel (z-direction) could be used to inspect the sample manually.

Images were acquired with a LabVIEW program (Flash Pathology B.V.), which had preset scanning programs with different fields of view and sampling densities, and therefore different acquisition times. The characteristics of the scanning programs are listed in Figure 1. The Fast Overview scan program was used to make an overview image of the entire sample. The High-Quality scan program was used to scan selected regions of interest (usually regions of 2 mm × 2 mm), often the cellular regions. Imaging time varied between 20 min and one hour and depended mostly on the surface area of the sample and the number of high-quality scans taken. The raw data contained signal intensities between 0 and 10,000 and were converted to 24-bit RGB images (8-bit for each color, i.e., intensities between 0 and 255). The signal intensities were scaled with a gamma correction in the LabVIEW program: γ = 0.5 for THG and SHG and γ = 0.7 for the 2PEF signal. The upper limit of the display range for each signal was manually determined to make sure that the cells and structures were well visible, without creating oversaturated images. Each signal is visualized in a different color: THG in green, SHG in red and 2PEF in blue. The images are a mixture of these colors. For example, collagen could generate SHG and THG, which resulted in an orange or yellow appearance in the HHGM images.

### 2.4. Pathologists’ Assessment

#### 2.4.1. Slide Score

To investigate the ability of the pathologists to interpret the HHGM images, a study was set up using Slide Score (https://www.slidescore.com, Amsterdam, The Netherlands, accessed on 31 January 2024), an online pathology platform which integrates a pathology viewer with a question sheet. The pathology viewer can be used to zoom in on the images and navigate to analyze the large images in detail. In addition, pathologists were able to enhance the colors of the images or (de)select different color channels.

#### 2.4.2. Training of the Pathologists

The pathologists were trained with a training session in which the technique was explained and some examples were given. The pathologists received a document containing the same explanation and examples as in the training session, which they were allowed to use during the assessment. To prevent any bias, only general examples were given (i.e., adipose tissue, cartilage, muscle, skin, immune cells, artifacts, etc.) and small (200 µm × 200 µm) close-ups of normal histopathological features—glomeruli, tubules, and a blood vessel—and even smaller close-ups (100 µm × 100 µm) of psammoma bodies, hyaline structures, spindle cells, blastema, epithelium, stroma and rhabdomyoblasts. These images were rotated and/or mirrored and changed in color intensity so that they could not be recognized in the assessment. Furthermore, the Slide Score study started with a training set, in which HHGM images and corresponding histology images from two excluded kidney samples (non-relevant diseases) were shown simultaneously. This training case was used to practice with the pathology viewer and the question sheet, and to see how the HHGM images corresponded with the histology images.

#### 2.4.3. Study Design

To mimic the normal workflow of the pathologist, limited case information was provided (age, gender and organ), and questions were based on how histology sections are usually evaluated (Appendix A). The order of the cases was the same for each pathologist so that they all had the same learning effect (Appendix A). For each case, pathologists first assessed the HHGM part (overview image of the entire sample and the high-quality close-ups). After the HHGM part, the pathologists received the corresponding H&E section with the same question sheet and could not see nor change their assessment of the HHGM part.

#### 2.4.4. Data Analysis

The consensus, determined by the majority opinion (≥6 of the pathologists), was used to calculate the diagnostic characteristics (sensitivity, specificity, positive predictive value, negative predictive value and accuracy). A method similar to MedCalc was used to calculate 95% confidence intervals [18]. Clopper–Pearson confidence intervals were used for sensitivity, specificity and accuracy. Standard logit confidence intervals given by Mercaldo et al. [19] were used for PPV and NPV, except for the values that reached 0% or 100%, which were calculated with Clopper–Pearson confidence intervals. Diagnostic characteristics (sensitivity, specificity, positive predictive value, negative predictive value and accuracy) were determined for two distinctions: normal versus abnormal, and non-tumor versus tumor. For both distinctions, the interobserver agreement was calculated using Krippendorff’s alpha with 95% confidence intervals and 1000 bootstrap iterations [20,21]. Generally, a Krippendorff’s alpha equal to or above 0.8 is considered a satisfactory level of agreement.

## 3. Results

Twenty-nine samples were obtained from eighteen patients with a renal tumor, sixteen with Wilms, one with renal cell carcinoma and one with congenital mesoblastic nephroma (Appendix A). The complete results of the pathologists’ assessment can be found in the Appendix A (Appendix A: assessment of normal versus abnormal tissue, Appendix A: statistical analysis of normal versus abnormal tissue, Appendix A: assessment of non-tumor versus tumor tissue, Appendix A: statistical analysis of non-tumor versus tumor tissue, Appendix A: assessment of tumor percentages, Appendix A: assessment of tumor diagnosis, Appendix A: assessment of Wilms tumor components, Appendix A: degree of certainty and correspondence between HHGM and H&E).

### 3.1. Diagnostic Characteristics

As shown in Table 1A, all diagnostic characteristics reached 100% for the distinction between normal and abnormal. The classifications and the diagnostic characteristics per pathologist can be found in the Appendix A, respectively. These tables show that there is some variation between HHGM and H&E diagnosis, and also between pathologists. Most of the discrepancies between HHGM and H&E per pathologist occurred in the same single case, which was reactive and had significant variability between pathologists on HHGM as well as H&E.

The distinction between non-tumors and tumors yielded a sensitivity of 100%, specificity of 91%, PPV of 95%, NPV of 100% and accuracy of 97%, as shown in Table 1B. The classifications and the diagnostic characteristics per pathologist can be found in the Appendix A, respectively. In addition, the pathologists assessed the percentage of viable tumors for each sample that they classified as tumors (Appendix A). These Appendix A show that most discrepancies between HHGM and H&E, and between pathologists, were in cases that were reactive or necrotic. The interobserver variability for HHGM and H&E is shown in Table 1C. For H&E, both distinctions have a Krippendorff’s alpha higher than 0.8 and are considered satisfactory. For HHGM, this applied to the distinction between normal and abnormal.

### 3.2. Comparison of HHGM and H&E for Normal Kidney

Pathologists were able to distinguish between a normal kidney and abnormal kidney on HHGM images with an accuracy of 100%. Figure 2 shows an HHGM image of a normal renal cortex, the corresponding histology image and the separate HHGM color channels with enhanced intensities. The HHGM images depict the normal renal architecture in a way very similar to histology. SHG (red) visualizes the collagen network surrounding the tubules and the glomeruli. THG (green) visualizes cells and structures, including the collagen fibers, which therefore appear orange in the HHGM images. The 2PEF (blue) signal is weak, but the enhanced 2PEF image shows that the proximal tubule cells have a stronger 2PEF signal than the distal tubule cells. In addition, the proximal tubule cells have a more granular appearance in the HHGM images than the distal tubule cells. There seems to be some SHG signal in the tubule cells as well; however, this is probably due to autofluorescence emission in the SHG detection wavelength range.

### 3.3. Diagnostic Characteristics for Tumor Diagnosis

The HHGM consensus and the H&E consensus were compared to the gold standard diagnosis to calculate the diagnostic accuracy. Cases that contained <33% tumor (based on the H&E mean tumor percentage in the Appendix A) were excluded from this calculation, leaving 18 relevant tumor samples. Table 2 shows that the diagnostic accuracy was 94% for both HHGM and H&E, respectively, as compared to the gold standard diagnosis. The tumor diagnosis per pathologist can be found in the Appendix A.

### 3.4. Comparison of HHGM and H&E for Renal Tumors

Pathologists were able to distinguish several renal tumor types on the HHGM images. Figure 3 shows the HHGM images and corresponding histology for several histopathological features of renal tumors. Overall, the HHGM images show good correspondence with the H&E images regarding tissue architecture. Cellular regions appear mostly green, while fibrotic regions are mostly red or orange. The three components in Wilms tumors (blastema, epithelium, stroma) can be recognized in the HHGM images; however, blastema and primitive epithelium might be difficult to distinguish, as they both appear green in the HHGM images. Foamy macrophages are prominent in the HHGM images since the granular cytoplasm creates a high THG signal. Pigments such as hemosiderin correlated with a high intensity in the 2PEF signal and are more blue or purple. Necrotic areas do not have a clear architecture in the HHGM images. Renal cell carcinomas show a papillary architecture with hyaline structures, cells with a prominent nucleolus, and psammoma bodies with a clear double border. Congenital mesoblastic nephromas have more elongated (spindle) cells, and the orientation of the collagen fibers reveals the fascicular pattern. Cellular areas and entrapped tubules can be recognized in the HHGM images as well.

### 3.5. Classification of Viable Wilms Tumor Components

Pathologists assessed the percentages of blastema, epithelium and stroma for each tumor sample that they classified as Wilms tumor (Appendix A). These assessments were divided into four categories: >66% blastema, >66% epithelium, >66% stroma and ‘mixed’ if none of the components were >66%. To calculate the agreement between HHGM and H&E, the majority opinion of the main component was determined for each sample, based on HHGM and H&E. To focus on the assessment of the viable components, Wilms tumor samples that mainly showed necrosis or reactive tissue were intentionally omitted in this analysis (based on the H&E mean vital tumor percentage, Appendix A, <33% tumor was considered irrelevant). Of the twenty-nine samples, nineteen samples were obtained from a confirmed Wilms tumor and four of those samples were mainly necrotic or reactive, leaving fifteen relevant Wilms tumor samples. Table 3 shows that 87% (13/15) of the cases have the same main component based on HHGM as on H&E. The two cases that were differently interpreted were of a similar nature (mixed versus epithelial).

### 3.6. Rhabdomyoblastic Differentiation and Anaplasia

Pathologists could indicate whether they found rhabdomyoblastic differentiation or anaplasia in the cases they classified as Wilms tumor. Rhabdomyoblasts can be recognized as a round structure in the SHG channel of the HHGM images (Figure 4), which was also explained in the training document for the pathologists. One case (case 06) consisted almost completely of stroma and rhabdomyoblasts. Indeed, five out of six pathologists who classified this case as Wilms tumor on the HHGM images identified the rhabdomyoblastic differentiation. One sample in this study contained anaplasia. Of the ten pathologists, six identified the anaplasia on the H&E, but none recognized it on the HHGM images. The cases where the pathologists assigned anaplasia on the HHGM images were often misinterpretations of hyaline globules or calcifications.

## 4. Discussion

This study presents the results of a multi-observer analysis conducted by expert pathologists to assess higher-harmonic-generation microscopy images of pediatric renal tumors, as compared to conventional histology. Pathologists accurately distinguished between normal and abnormal tissue on HHGM images with 100% agreement (29/29), and identified tumor versus non-tumor tissue with 97% accuracy (28/29). The interobserver agreement for HHGM was satisfactory for the distinction between normal and abnormal, but poor for the distinction between non-tumor and tumor. Discrepancies between HHGM and H&E diagnoses were mainly observed in samples that were necrotic or reactive. Additionally, pathologists successfully identified key structures, such as glomeruli and tubules in normal renal tissue, and blastema, epithelium and stroma in Wilms tumors, demonstrating that HHGM can be a valuable tool in applications where detailed tissue architecture is critical. For instance, it could be used to assess non-tumoral renal biopsies in nephropathology where a minimum of 15 glomeruli are required for analysis [22], or to confirm tissue viability and tumoral representativity in tumoral specimens, offering a non-tissue-consuming alternative for frozen section analysis for molecular analysis.

Our results indicate that conventional histological assessment can exhibit significant interobserver variability. In one case, the estimated viable tumor percentages varied from 10% to 100%. In a Wilms tumor case predominantly composed of blastema and epithelium, estimates ranged from 100% epithelium to 85% blastema. In two other Wilms tumor cases featuring a mix of epithelium and stroma, the estimates varied from over 80% epithelial to over 80% stromal. Since the HHGM images have a clear distinction between cells (green) and collagen (red), an automated analysis of HHGM images might have potential for an objective quantification of the Wilms tumor components.

This study has several limitations. HHGM was performed on freshly excised tissue fragments, resulting in a limited number of samples, most of which were from Wilms tumors. Other tumor types, such as CMN and RCC, were represented only once or twice, while rhabdoid tumors were not represented at all. Therefore, we have too little data to draw conclusions about the diagnostic accuracy of HHGM for tumor diagnosis. To assess the diagnostic value of HHGM, diagnoses other than tumors should also be included. With regard to samples, we were limited to the resection specimens in the Princess Maxima Center, which is a pediatric oncology center, with only occasional non-tumor cases.

Additionally, tissue samples were selected based on macroscopic examination, which meant they were not always representative of tumors. Some samples were reactive, necrotic or stained with Indian ink, which caused artifacts in the HHGM images. The heterogeneity of the Wilms tumor samples also affected their recognition, with some pathologists classifying the tumor type as uncertain. They noted that additional staining, immunohistochemistry or more samples would be necessary to rule out other diagnoses.

Some HHGM images contained artifacts, especially bright speckles caused by Indian ink, which was used to indicate surgical margins. Other artifacts arose from the set-up, including air bubbles in the immersion oil or beneath the sample, which create shadow-like effects on the images. These artifacts usually only affect part of the image (usually shadows/dark areas of less than 100 × 100 microns), and therefore have limited influence on the diagnostic accuracy. Similar to histology, it is important to be aware that these artifacts exist and how to interpret them.

To simulate a pathologist’s normal workflow, we provided the patient’s age and gender, and the organ involved, which may have introduced bias into the tumor diagnosis. Furthermore, the variability in sample representativity and interobserver differences made it challenging to calculate precise diagnostic characteristics. As a result, these characteristics were based on consensus, defined as the majority opinion of ≥6 pathologists, and were focused on two distinctions: normal vs. abnormal, and tumor vs. non-tumor.

Pathologists had limited training in interpreting the HHGM images, which led to some misinterpretations. For example, some pathologists mistakenly identified calcifications and hyaline globules as signs of anaplasia, or misclassified a lymph node with RCC metastasis as normal or reactive, despite it being completely infiltrated with tumor. These errors highlight the fact that interpreting HHGM images is not always intuitive and further training is needed to accurately identify tumor features and metastases.

In this study, pathologists first assessed the HHGM image of each case, and then the corresponding H&E section, to create an extra learning effect during the assessment and to be able to receive feedback on the correspondence between HHGM and H&E and possible misinterpretations. This design could have created a bias in the H&E assessment, because pathologists could have based their decisions on both HHGM and H&E. However, we assumed that pathologists would prioritize the H&E assessment, because histology is their gold standard and they are very experienced, while the HHGM technique was new to them. During this study, we only compared HHGM with conventional histology, using H&E sections. Another improvement would be to compare the HHGM images with other histopathological techniques, for example immunohistochemistry, to determine which other histopathological features can be observed in HHGM images.

The imaging time in our study ranged from 30 to 60 min, depending on the size of the sample and the number of high-quality scans. A quick overview scan (10 s per mm^2^) was sufficient for identifying tissue architecture, glomeruli, tubules and cellular areas. Planned technical advancements are expected to increase imaging speed by a factor of four, making HHGM an even faster intraoperative alternative.

## 5. Conclusions

Pathologists were able to distinguish the architectural differences between normal renal tissue and tumor tissue in HHGM images. The accuracy of HHGM in differentiating normal from abnormal tissue was 100% (95% confidence interval: 88–100%) and in distinguishing non-tumor versus tumor tissue, it was 97% (95% confidence interval: 82–100%). These findings demonstrate that HHGM is a reliable technique for assessing the representativity of renal tissue samples.

## Figures and Tables

**Figure 1 cancers-17-01693-f001:**
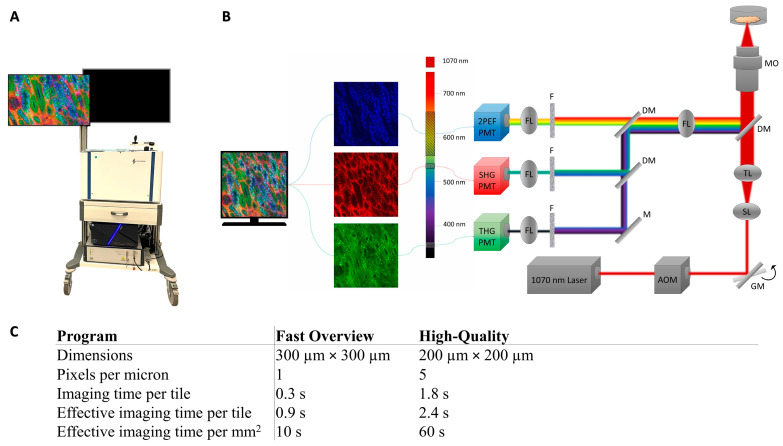
Image of the transportable HHG microscope (**A**) and schematic set-up (**B**) used in this research. The set-up includes a near-infrared laser (1060 nm), an acousto-optic modulator (AOM), galvanometer mirrors (GMs), scan lens (SL), tube lens (TL), dichroic mirrors (DMs), microscope objective (MO), focus lens (FL), mirror (M), filters (F), and photo-multiplier tubes (PMTs). (**C**) Characteristics of the used scanning programs. The ‘Effective imaging time’ also includes the movement of the XYZ stage, which took around 0.6 s per image.

**Figure 2 cancers-17-01693-f002:**
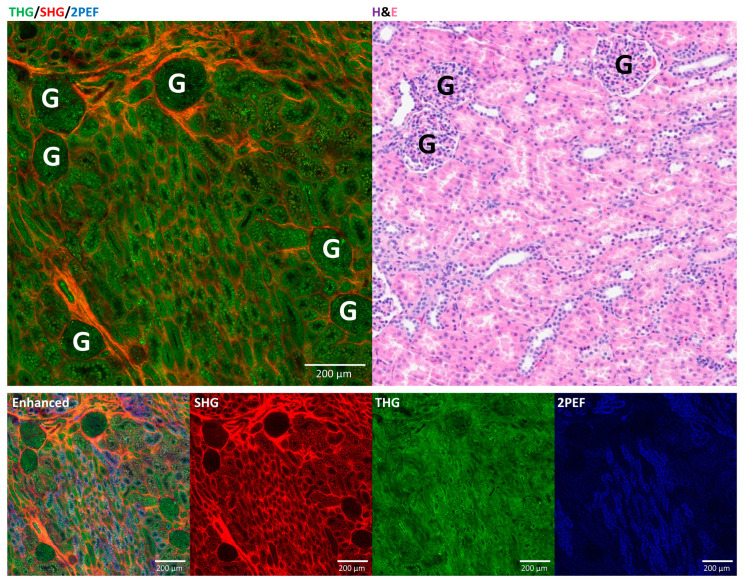
Comparison of HHGM images and H&E of normal kidney cortex, showing glomeruli (G) and tubules, and analysis of separate signals (SHG, THG, 2PEF) with increased intensity. THG visualizes cells and structures, while SHG and 2PEF visualize collagen and autofluorescence, respectively.

**Figure 3 cancers-17-01693-f003:**
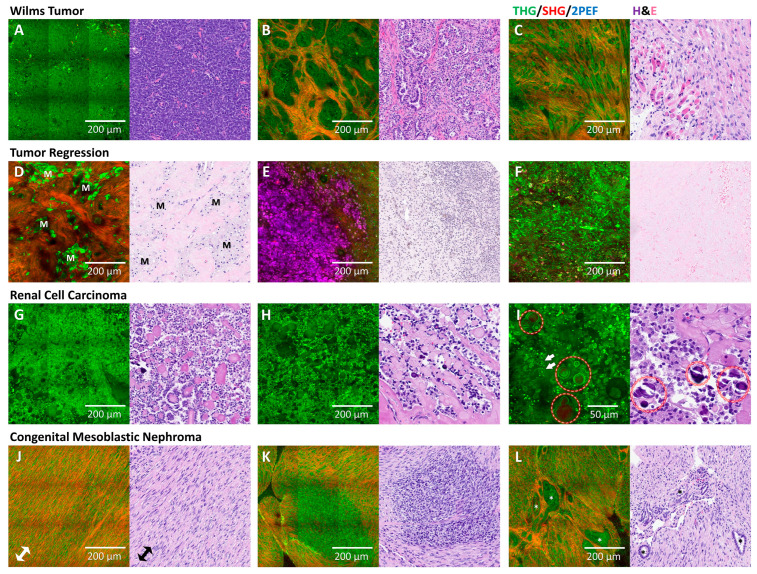
Comparison of HHGM images and H&E for Wilms tumor, renal cell carcinoma and congenital mesoblastic nephroma. Wilms tumor is a heterogeneous tumor and consists of three different components. The blastemal component (**A**) is cellular and is mostly visualized in the THG channel. The epithelial component (**B**) shows cells arranged in tubular structures. The stromal component (**C**) may have rhabdomyoblastic differentiation as shown in this case; the rhabdomyoblasts are round and have a large cytoplasm, and the more differentiated rhabdomyoblasts are elongated and sometimes have a visibly striated sarcoplasm, which is better shown in Figure 4. Examples of tumor regression (**D**–**F**) show dense fibrosis and foamy macrophages (annotated as M), which are prominent in the HHGM image (**D**); hemosiderin pigments create a strong 2PEF signal in the HHGM images (**E**); necrosis shows unrecognizable structures with no clear tissue architecture (**F**). Renal cell carcinoma shows some papillary arrangement (**G**) and hyaline structures (**H**), and a close-up (**I**) shows psammoma bodies (circled) and cells with a prominent nucleolus (arrows). Congenital mesoblastic nephroma shows fascicular patterns of spindle-shaped cells (in (**J**), orientation of the bundles is indicated with an arrow), cellular areas (**K**) and entrapped renal tubules ((**L**), annotated with *).

**Figure 4 cancers-17-01693-f004:**
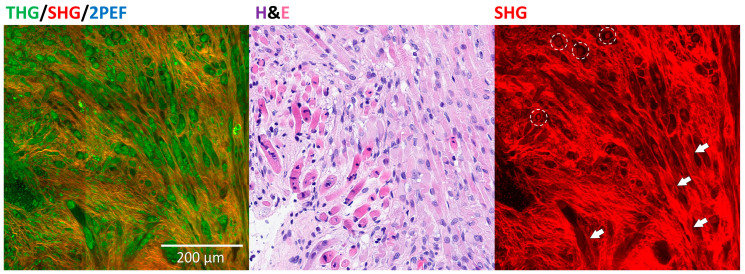
Wilms tumor with rhabdomyoblastic differentiation. The rhabdomyoblasts (circled) and striated muscle cells (arrows) create the SHG signal.

**Table 1 cancers-17-01693-t001:** Classification of cases in abnormal versus normal (**A**), and tumor versus non-tumor (**B**), based on the HHGM consensus and the H&E consensus. Interobserver agreement for HHGM and H&E is calculated with Krippendorff’s alpha (**C**). Additionally, 95% confidence intervals are given in square brackets. * In one case, the H&E consensus was indeterminable (equal amount of normal and abnormal); this case was considered abnormal. ** In two cases, the HHGM consensus was indeterminable (equal amount of non-tumor and tumor); these cases were considered as non-tumor.

A				
Normal vs. Abnormal		H&E Consensus		
		Abnormal	Normal	
HHGM Consensus	Abnormal	23 *	0	PPV = 100% [85–100%]
	Normal	0	6	NPV = 100% [54–100%]
		Sensitivity = 100% [85–100%]	Specificity = 100% [54–100%]	Accuracy = 100% [88–100%]
B				
Non-Tumor vs. Tumor		H&E Consensus		
		Tumor	Non-Tumor	
HHGM Consensus	Tumor	18	1	PPV = 95% [74–99%]
	Non-Tumor	0	10 **	NPV = 100% [69–100%]
		Sensitivity = 100% [81–100%]	Specificity = 91% [59–100%]	Accuracy = 97% [82–100%]
C				
Krippendorff’s Alpha [95% confidence interval, 1000 bootstrap iterations]			HHGM	H&E
Classification Normal versus Abnormal			0.82 [0.66, 0.93]	0.89 [0.72, 1.00]
Classification Non-Tumor versus Tumor			0.45 [0.25, 0.59]	0.81 [0.64, 0.92]

**Table 2 cancers-17-01693-t002:** HHGM consensus and H&E consensus versus the gold standard diagnosis of the tumor types. The 95% confidence intervals are given in square brackets. * Renal cell carcinoma metastasis in a lymph node, also the first case with RCC in the assessment, with no clear consensus between Wilms tumor and renal cell carcinoma. ** Wilms tumor sample with many cysts and calcifications, and no clear consensus between Wilms tumor, renal cell carcinoma and metanephric adenoma.

		Gold Standard Diagnosis		
		CMN	RCC	WT
HHGM Consensus	CMN	1	0	0
	RCC	0	1	0
	WT	0	0	15
	No consensus	0	1 *	0
		**Gold Standard Diagnosis**		
		**CMN**	**RCC**	**WT**
H&E Consensus	CMN	1	0	0
	RCC	0	2	0
	WT	0	0	14
	No consensus	0	0	1 **

HHGM accuracy for Wilms tumor versus non-Wilms tumor: 94% (17/18) [73–100%]; H&E accuracy for Wilms tumor versus non-Wilms tumor: 94% (17/18) [73–100%].

**Table 3 cancers-17-01693-t003:** Classification of main Wilms tumor components: HHGM majority opinion versus H&E majority opinion.

		H&E Majority Opinion			
		>66% Blastema	>66% Epithelium	>66% Stroma	Mixed
HHGM Majority Opinion	>66% Blastema	2	0	0	0
	>66% Epithelium	0	0	0	0
	>66% Stroma	0	0	7	0
	Mixed	0	2	0	4
Agreement: 87% (13/15).					

## Data Availability

The datasets generated during and/or analyzed during the current study are available from the corresponding author on reasonable request.

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
