# Peer review of "Multi-Observer Study on Diagnostic Accuracy of Pediatric Renal Tumors Imaged with Higher-Harmonic-Generation Microscopy"

_cancers, 2025, doi:10.3390/cancers17101693_

Round 1
Reviewer 1 Report
Comments and Suggestions for Authors
The submitted manuscript addresses a novel imaging modality for evaluation of pediatric renal tumors, primarily Wilms tumor. This study is significant a rapid intraoperative alternative to conventional histology, which could revolutionize pediatric renal tumor diagnosis and intraoperative management. The clinical implications are substantial, particularly in contexts such as biopsy evaluation and biobanking, where rapid, accurate, and non-destructive methods are advantageous.
- The methodology requires clarification regarding inter-observer variability and training protocols.
- Tumor representation was skewed towards Wilms tumor, with insufficient representation of rarer pediatric renal tumors, limiting the generalizability of the findings.
- The conclusions drawn are robust for the stated diagnostic distinctions; however, clarifications regarding sample selection bias, limitations in training pathologists on HHGM interpretation, and the impact of artifacts on diagnostic accuracy should be addressed.
Reviewer 2 Report
Comments and Suggestions for Authors
This study evaluates higher harmonic generation microscopy (HHGM) for distinguishing between normal and abnormal renal tissue and pediatric renal tumors. By analyzing twenty-nine samples from eighteen patients, pathologists achieved 100% accuracy in differentiating normal from abnormal tissue and 97% accuracy in tumor identification using HHGM. The technique provides fast, label-free imaging that closely aligns with conventional histology, making it a promising tool for real-time assessment and intraoperative feedback in pediatric renal diagnostics. However, there are some limitations related to sample size and pathologist training.
For label-free imaging of tissue, the optimized excitation wavelength for cells should ideally range from 500 to 700 nm. For instance, this range can effectively visualize proteins, amino acids, NADPH, and certain organelles. It would be beneficial to explain the decision to choose only 1070 nm and to compare it with the use of lower-wavelength excitation sources to determine if they yield more informative images.
Although the results reflect an excellent correlation, the pathologists had limited training in interpreting HHGM images. This limitation could lead to variability in their assessments and potential misinterpretation of features when evaluating a larger dataset.
In section "2.4.3. Study Design" on page 6, it should be noted that when pathologists evaluate the corresponding H&E sections, they are basing their decisions on both HHGM and H&E findings. The results from H&E may be influenced by their previous evaluations.
Additionally, incorporating other histopathological techniques, such as immunohistochemistry or molecular profiling, could further clarify the clinical relevance of HHGM findings.
There are also some reference errors in the tables.
It would be great to highlight some of the important features in Figure 2 and 3 using arrows and circles to help readers to better understand.
Reviewer 3 Report
Comments and Suggestions for Authors
The authors use higher harmonic generation microscopy (HHGM) method to visualize the renal tissue, which prodives comparable information with traditional method. With the images from HHGM, pathologists achieved high accuracy in distinguishing between normal renal tissue, abnormal renal tissue and renal tumors. The method proposed by the author can effectively improve the efficiency The sample size is small, while the result is still persuasive.
- Many references show errors, e.g. Section Paragraph 1,2.3 Line1, paragraph 2, line4 etc.
- In section 2.2, authors proposed an experi-enced pathologist, who did not participate in the HHGM assessment, selected (vital) tu-mor and normal samples for HHGM imaging, based on macroscopic evaluation. From a statistical point of view, it is difficult for a researcher to select a sample without bias. If two different researchers can select samples separately and analyze different results, the conclusion will be more credible.
- In section 2.3, the authors metioned that the upper limit of the display range for signal was manually determined, the authors can introduce the metric in detail, because it is important to reproduce the results for other researchers.
- In section 2.4.4, the authors use voting method to decide the result. Can authors use models like random forest so that all information to make decision?
Round 2
Reviewer 1 Report
Comments and Suggestions for Authors
Authors successfully addressed my comments.
Reviewer 3 Report
Comments and Suggestions for Authors
- There are errors in the references cited in the article. Please use relevant software such as endnote or other softwares to correct the bugs.
- Other comments have been addressed.